# Ameloblastomas Exhibit Stem Cell Potential, Possess Neurotrophic Properties, and Establish Connections with Trigeminal Neurons

**DOI:** 10.3390/cells9030644

**Published:** 2020-03-06

**Authors:** Pierfrancesco Pagella, Javier Catón, Christian T. Meisel, Thimios A. Mitsiadis

**Affiliations:** 1Orofacial Development and Regeneration, Institute of Oral Biology, Center of Dental Medicine, University of Zurich, 8032 Zurich, Switzerland; christian.meisel@zzm.uzh.ch; 2Department of Anatomy and Embryology, Faculty of Medicine, University Complutense Madrid, 28040 Madrid, Spain; javicaton@med.ucm.es

**Keywords:** ameloblastoma, oral cancers, cancer stem cells, SOX2, innervation, neurotrophins, notch signaling, trigeminal neurons, neuronal contacts, microfluidic devices

## Abstract

Ameloblastomas are locally invasive and aggressive odontogenic tumors treated via surgical resection, which results in facial deformity and significant morbidity. Few studies have addressed the cellular and molecular events of ameloblastoma onset and progression, thus hampering the development of non-invasive therapeutic approaches. Tumorigenesis is driven by a plethora of factors, among which innervation has been long neglected. Recent findings have shown that innervation directly promotes tumor progression. On this basis, we investigated the molecular characteristics and neurotrophic properties of human ameloblastomas. Our results showed that ameloblastomas express dental epithelial stem cell markers, as well as components of the Notch signaling pathway, indicating persistence of stemness. We demonstrated that ameloblastomas express classical stem cell markers, exhibit stem cell potential, and form spheres. These tumors express also molecules of the Notch signaling pathway, fundamental for stem cells and their fate. Additionally, we showed that ameloblastomas express the neurotrophic factors NGF and BDNF, as well as their receptors TRKA, TRKB, and P75/NGFR, which are responsible for their innervation by trigeminal axons in vivo. In vitro studies using microfluidic devices showed that ameloblastoma cells attract and form connections with these nerves. Innervation of ameloblastomas might play a key role in the onset of this malignancy and might represent a promising target for non-invasive pharmacological interventions.

## 1. Introduction

Ameloblastoma is a locally invasive and aggressive odontogenic neoplasia accounting for approximately 10% of tumors affecting the mandible and the maxilla [1]. Ameloblastoma is the most common benign intraosseous epithelial odontogenic tumor. Radiographically it appears as a multilocular or bubble-like radiolucency. Eighty percent of all ameloblastomas are located in the mandible and generally in the posterior portion with some relationship to unerupted teeth. Ameloblastomas can be classified in three main histopathological subtypes, namely ameloblastoma, unicystic ameloblastoma, and extraosseous/peripheral ameloblastoma [2]. Ameloblastomas grow slowly, are locally invasive and have a high recurrence rate [3]. The treatment of choice generally consists of jaw resection, which is often associated with significant morbidity. Only a few studies have addressed the molecular mechanisms underlying the onset and progression of ameloblastomas, thus hampering the development of non-invasive therapies [3]. It has been shown that ameloblastomas are often characterized by overexpression of epidermal growth factor receptor (EGFR) and ERBB, as well as by mutations in *SMO* (smoothened), *BRAF*, and members of the MAPK pathways [3,4,5,6]. These mutations, isolated or combined, often confer these cancers drug resistance [3,6,7]. It was recently shown that ameloblastomas express high levels of the dental epithelial stem cell marker SOX2, suggesting that these tumors might originate from SOX2-expressing cells from the dental lamina [8]. SOX2 is a key promoter of stem cell self-renewal, and its sustained expression might drive ameloblastoma pathogenesis [8]. Different studies have reported expression of Notch signaling receptors and ligands in ameloblastomas [9,10,11], suggesting its implication in ameloblastoma onset and progression. Notch signaling is a highly conserved pathway that acts as a hub of the various signaling networks, and is a major driver of stem cell fate determination in development, homeostasis, and disease [12,13,14]. While many studies support a pivotal role for Notch signaling in the pathogenesis of other oral cancer types [15,16,17,18,19], little information exists concerning its role in ameloblastomas.

In the last few decades, increasing evidence indicates that innervation plays an active role in cancer onset, progression, and metastatization [20,21,22,23,24,25,26,27]. Many classes of cancer are infiltrated by nerve fibers, and their presence is associated with poor prognosis [21,22,23]. Indeed, experiments conducted on mice have shown that denervation of primary tumors results in a strong decrease in tumor growth and dissemination. These observations were first conducted on prostate cancers [25], but in recent years nerve dependence has been demonstrated for many classes of tumors, including gastric cancers [27], melanomas [20], pancreatic cancers [24,26], and head and neck squamous cell carcinomas [22,23]. In addition to driving cancer progression, nerve fibers are the mediators of cancer-associated pain [28,29].

To date, no study has investigated the relationship between innervation and ameloblastomas, and the basis for ameloblastoma-induced pain [30]. This neglected aspect could be of paramount relevance, as nerve-derived signals might directly modulate ameloblastoma onset and progression, as already demonstrated in other cancer types.

Innovative, state-of-the-art culture systems are actively used to analyze cell–cell interactions. Among these, microfluidic devices emulate complex cell interactions, thus recapitulating and mimicking multicellular architecture, cell–cell interactions, and physical microenvironment of functional units of living organs [31,32]. These miniaturized models, also called organs-on-chips, allow precise control of the secreted signaling molecules and provide a real-time readout on basal cell functions that permits the reconstitution of organ physiology.

In this work we characterized the expression of stem cell markers, Notch receptors and ligands that might affect the stem cell fate, and of neurotrophins in ameloblastomas. We then assessed the neurotrophic effects of ameloblastoma cells on trigeminal ganglia using microfluidic devices. The results show that ameloblastomas express stem cell markers, molecules of the Notch signaling pathway and neurotrophins. Ameloblastomas are innervated in vivo, and this was shown in microfluidic systems in vitro, where ameloblastoma cells attracted and established contacts with trigeminal axons.

## 2. Materials and Methods

### 2.1. Collection of Human Material

Patient tissue samples and data were provided by Professor Peter Morgan (King’s College London, London, UK; NHS) from Guy’s & St Thomas’ Head & Neck Biobank—part of the KHP Cancer Biobank, which is supported by the Department of Health via the National Institute for Health Research (NIHR) comprehensive Biomedical Research Centre award and Guy’s & St Thomas’ NHS Foundation Trust (REC 12-EE-0493; REC10/h0703/54). All patients provided their informed consent. Immunostaining were performed on a plexiform ameloblastoma isolate from a male patient, 16 years old (enucleation from the mandible, 48 mm × 45 mm × 28 mm), and a basal-cell type ameloblastoma isolated from a male patient, 73 years old (mandibular resection, 45 mm × 28 mm × 22 mm).

AB10 cells were isolated from a primary solid multi-cystic ameloblastoma resected from the right/posterior mandible, as previously described [6,33]. Briefly, fresh samples from ameloblastoma tumors were cut into small pieces (approximately 1 mm × 1 mm) and placed in T25 cell culture flasks with 1 mL CnT-24 medium (CELLnTEC), supplemented with Pen/Strep/Amphotericin B Solution (CELLnTEC). Outgrowing ameloblastoma cells were harvested at confluence and maintained in the CnT-24 medium [6,33].

For sphere formation, AB10 cells were cultured in Keratinocyte Serum Free Medium (KSFM) supplemented with EGF and Bovine Pituitary Extract (BPE) and 1% Penicillin/Streptomycin. For each well (12 well/plate), 20,000 cells were resuspended in 50 μL of medium. 50 μL of Matrigel (354230, Corning) were then added to the cell suspension, the mixture plated in a 12 well plate and incubated at 37 °C for 1 h 1 mL of KSFM was then added [34].

### 2.2. Immunostaining on Tissue Sections

Immunohistochemistry on paraffin sections was performed as described previously [35]. Briefly, paraffin blocks were cut as 5 μm-thick sections. The sections were rehydrated by incubation in Xylol followed by a series of ethanol solutions (100%–30%) and distilled H_2_O. Endogenous peroxidases were inhibited by incubating the sections in a solution composed of 3% H_2_O_2_ in Methanol at −20 °C for 20 min. Specimens were then blocked with PBS supplemented with 2% fetal bovine serum and thereafter incubated with primary antibodies for 1 h at room temperature. The following primary antibodies were used: Rabbit anti-β-III-tubulin (1:100, ab18207, Abcam, Cambridge, UK); Rabbit anti-Keratin 14 (1:200, 905301, BiolLegend, San Diego, CA, USA), and rabbit pAb anti-PGP 9.5 (1:100, GTX109637, GeneTex). For negative controls, the primary antibody was omitted. The sections were then incubated with appropriate secondary antibodies (Vector Vectastain ABC kit PK-4001-1; Vector Laboratories LTD, Peterborough, UK). Sections were then incubated with AEC (3-amino-9-ethylcarbazole; AEC HRP substrate kit-SK4200; Vector Laboratories LTD, Peterborough, UK) to reveal the staining, counterstained with Toluidine Blue, mounted with Glycergel (C0563, Agilent Technologies, Santa Clara, CA, USA) and finally imaged with a Leica DM6000 light microscope (Leica Microsystems, Schweiz AG, Heerbrugg, Switzerland).

For double immunofluorescent staining the endogenous peroxidase inhibition step was omitted. Primary antibodies were applied simultaneously for 1 h at room temperature. Sections were then incubated with fluorochrome-conjugated secondary antibodies for 1 h at room temperature and in the dark. The following primary antibodies were used: Goat anti-NGF (1:100, AF256NA, R&D Systems, Minneapolis, MN, USA), Rabbit anti-BDNF (1:100, ab108319-100UL, Abcam, Cambridge, UK), mouse IgG1 α-neurofilament (α-NF) antibody (1:100, Hybridoma Bank, Iowa City, IA, USA), rabbit mAb anti-neurofilament (1:200, Cell Signaling Technology, 2837), mouse mAb anti-Vimentin (1:100, DAKO, M0725), rabbit mAb anti-TrkA (1:50, Abcam, ab76291), rabbit pAb anti-TrkA (1:100, Sigma-Aldrich, 06-574), rabbit pAb anti-TrkB (1:100, R&D Systems, AF1494), rabbit pAb anti-TrkC (1:100, R&D Systems, AF1404), rabbit mAb anti-nerve growth factor receptor (NGFR/p75; 1:100, Sigma-Aldrich, N3908), rabbit pAb anti-β-III-Tubulin (1:200, Abcam, ab18207), rabbit pAb anti-Synapsin I (1:100, Abcam, ab64581), rabbit mAb anti-wide spectrum Cytokeratins (1:100, Abcam, ab9377), rabbit pAb anti-Notch1, Notch2, Notch3 [35], rabbit pAb anti-Dll1 (1:100, ab1054, Abcam), rabbit pAb anti-Dll4 (1:50, ab7280, Abcam), and rabbit pAb anti-Jag1 (1:50, ab7771, Abcam). For negative controls, primary antibody was omitted. The following secondary antibodies were used: Alexa-568 Donkey anti-Rabbit (1:500-A-10042; Thermo Fisher,), Alexa-488 Chicken anti-Goat (1:500-A-21467; Thermo Fisher,), Alexa-488 Goat anti-Rabbit (1:500-A32731; Thermo Fisher), and Alexa-568 Goat anti-Rat (1:500-A-11077; Thermo Fisher,). DAPI (4′,6-Diamidino-2-Phenylindole-D1306; Thermo Fisher) was then used for nuclear staining. After immunofluorescent staining, samples were mounted in ProLong™ Diamond Antifade Mountant (P36965; Thermo Fisher,) and imaged with a Leica SP8 Inverted Confocal Laser Scanning Microscope (Leica Microsystems- Schweiz AG, Heerbrugg, Switzerland).

### 2.3. Gene Expression Analysis—Real Time PCR

Cells were collected by trypsinization, snap-frozen in liquid nitrogen and stored at −80 °C. RNA isolation on snap-frozen cells and TGG/DRG was performed with the RNeasy Plus Universal Mini Kit according to the instructions (Qiagen AG, Hombrechtikon ZH, Switzerland). Reverse transcription of the isolated RNA was performed using the iScript™ cDNA synthesis Kit and according to the instructions given (Bio-Rad Laboratories AG, Cressier FR, Switzerland). Briefly, 1000 ng of RNA were used for reverse transcription into cDNA. Nuclease-free water was added to produce a total of 15 μL, 4 μL of 5× iScript reaction mix, and 1 μL of iScript reverse transcriptase were added per sample in order to obtain a total volume of 20 μL. The reaction mix was then incubated for 5 min at 25 °C, for 30 min at 42 °C, and for 5 min at 85 °C using a Biometra TPersonal Thermocycler (Biometra AG, Göttingen, Germany).

Quantitative real-time PCR. The 3-step quantitative real-time PCRs were performed using an Eco Real-Time PCR System (Illumina Inc., San Diego CA, USA). Expression level analysis were carried out using the SYBR® Green PCR Master Mix (Applied Biosystems, Carlsbad CA, USA) in combination with specific oligonucleotide primers: *hNGF*, Fw: 5′- GGC AGA CCC GCA ACA TTA CT-3′, Rv: 5′- CAC CAC CGA CCT CGA AGT C-3′; *hBDNF*, Fw: 5′- GGC TTG ACA TCA TTG GCT GAC-3′, Rv: 5′- CAT TGG GCC GAA CTT TCT GGT-3′; *hGDNF*, Fw: 5′- AGC AGT GAC TCA AAT ATG CCA GA-3′, Rv: 5′- GCC TCT CCG ACC TTT TCC TC-3′; *hNT-3*, Fw, 5′- AAC GCG ATG TAA GGA AGC CA-3′, Rv: 5′- AGT GCT CGG ACG TAG GTT TG-3′; *hGAPDH*, Fw: 5′- AGG GCT GCT TTT AAC TCT GGT-3′, Rv: 5′- CCC CAC TTG ATT TTG GAG GGA-3; *hTP63*, Fw: 5′- GAA ACG TAC AGG CAA CAG CA-3′, Rv: 5′- GCT GCT GAG GGT TGA TAA GC-3′; *hBMI1*, Fw: 5′- CCA GGG CTT TTC AAA AAT GA-3′, Rv: 5′- CCG ATC CAA TCT GTT CTG GT-3′; *hCDH1*, Fw: 5′- TGC CCA GAA AAT GAA AAA GG-3′, Rv: 5′- GTG TAT GTG GCA ATG CGT TC-3′; *hACTA2*, Fw: 5′- ACC CAC AAT GTC CCC ATC TA-3′, Rv: 5′- GAA GGA ATA GCC ACG CTC AG-3′; *hFN1*, Fw: 5′- CCC AAC TGG CAT TGA CTT TT-3′, Rv: 5′- CTC GAG GTC TCC CAC TGA AG-3′; *hSOX2*, Fw: 5′- GCC GAG TGG AAA CTT TTG TCG-3′, Rv: 5′- GGC AGC GTG TAC TTA TCC TTC T-3′; hNOTCH1, Fw: 5′-GAG GCG TGG CAG ACT ATG C-3′, Rv: 5′-CTT GTA CTC CGT CAG CGT GA-3′; hNOTCH2, Fw: 5′-ACA GTT GTG TCT GCT CAC CAG GAT-3′, Rv: 5′-GCG GAA ACC ATT CAC ACC GTT GAT-3′; hNOTCH3, Fw: 5′-TGG CGA CCT CAC TTA CGA CT-3′, Rv: 5′-CAC TGG CAG TTA TAG GTG TTG AC-3′; hJAG1, Fw: 5′-GTC CAT GCA GAA CGT GAA CG-3′, Rv: 5′-GCG GGA CTG ATA CTC CTT GA-3′; hJAG2, Fw: 5′-TGG GCG GCA ACT CCT TCT A-3′, Rv: 5′-GCC TCC ACG ATG AGG GTA AA-3′; hDLL1, Fw: 5′-TGT GAC GAG TGT ATC CGC TAT-3′, Rv: 5′-GTG TGC AGT AGT TCA GGT CCT-3′; hDLL4, Fw: 5′-TGG GTC AGA ACT GGT TAT TGG A-3′, Rv: 5′-GTC ATT GCG CTT CTT GCA CAG-3′; hHES1, Fw: 5′-TCA ACA CGA CAC CGG ATA AAC-3′, Rv: 5′-GCC GCG AGC TAT CTT TCT TCA-3′.

The reaction mix was composed of 5 μL of SYBR® Green PCR Master Mix reverse and forward primers (200 nM), and 2 ng of template cDNA. The thermocycling conditions were 95 °C for 10 min, followed by 40 cycles of 95 °C for 15 s, 55 °C for 30 s, and 60 °C for 1 min. Melt curve analysis was performed at 95 °C for 15 s, 55 °C for 15 s, and 95 °C for 15 s. Expression levels were calculated by the comparative ΔCt method (2 − ΔCt formula), normalizing to the Ct-value of the *GAPDH* housekeeping gene.

### 2.4. Preparation of Microfluidic Devices

Microfluidic devices were prepared as previously described [36,37]. Glass coverslips were coated overnight at 37 °C with 0.1 mg/mL poly-D-lysine and stored in 70% Ethanol at 4 °C. Polydimethylsiloxane (PDMS) microfluidic devices (Millipore A150, Switzerland, 2 cm × 2 cm) were punched with a 1 mm-diameter biopsy punch on the neuronal side to enable the insertion of the trigeminal ganglion and then sterilized with 70% ethanol. Both glass coverslips and microfluidic devices were then left to dry completely under the laminar flow hood for approximately 2 h. In sterile conditions, glass coverslips were placed in a 6-wells plate. The microfluidic devices were then mounted onto the glass coverslips and pressed gently to ensure proper adhesion. After mounting, the microfluidic devices were coated with Laminin (5 μg/mL, in Neurobasal Medium) overnight at 37 °C. In order to prevent the persistence of air bubbles in the culture chambers, the coated microfluidic devices were placed under vacuum. After coating, the Laminin solution was removed, and the culture chambers filled with the appropriate culture medium.

### 2.5. Mouse Handling and Trigeminal Ganglia Dissection

All mice were maintained and handled according to the Swiss Animal Welfare Law and in compliance with the regulations of the Cantonal Veterinary Office, Zurich (License number: 151/2014; 146/2017). The animal facility provided standardized housing conditions, with a mean room temperature of 21 ± 1 °C, relative humidity of 50% ± 5%, and 15 complete changes of filtered air per hour (HEPA H 14filter); air pressure was controlled at 50 Pa. The light/dark cycle in the animal rooms was set to a 12 h/12 h cycle (lights on at 07:00, lights off at 19:00) with artificial light of approximately 40 Lux in the cage. The animals had unrestricted access to sterilized drinking water, and ad libitum access to a pelleted and extruded mouse diet in the food hopper (Kliba No. 3436; Provimi Kliba/Granovit AG, Kaiseraugst, Switzerland). Mice were housed in a barrier-protected specific pathogen-free unit and were kept in groups of max. 5 adult mice per cage in standard IVC cages (Allentown Mouse 500; 194 mm × 181 mm × 398 mm, floor area 500 cm^2^; Allentown, New Jersey, USA) with autoclaved dust-free poplar bedding (JRS GmbH + Co KG, Rosenberg, Germany). A standard cardboard house (Ketchum Manufacturing, Brockville, Canada) served as a shelter, and tissue papers were provided as nesting material. Additionally, crinklets (SAFE^®^ crinklets natural, JRS GmbH + Co KG, Rosenberg, Germany) were provided as enrichment and further nesting material. The specific pathogen-free status of the animals was monitored frequently and confirmed according to FELASA guidelines by a sentinel program. The mice were free of all viral, bacterial, and parasitic pathogens listed in FELASA recommendations [38].

C57/BL6J mice were time mated. Successful mating was assessed by a vaginal plug check, and the day of plug was considered as the day of embryonic development 0.5 (E0.5). Pregnant females were anesthetized with isoflurane and sacrificed by CO_2_ inhalation followed by decapitation to ensure death. Trigeminal ganglia were dissected from the embryonic day 14.5–16.5 (E14.5–E16.5) mouse embryos. Dissections were performed in cold Dulbecco’s phosphate buffered saline (PBS). Dissected ganglia were preserved in PBS, on ice, until culture. *N* = 3 embryos were used for the experiment.

Trigeminal ganglia were then cultured in Neurobasal Medium (Gibco) supplemented with B27 (Gibco 17504-044), GlutaMAX, Penicillin/Streptomycin (10 U/mL), 50 ng/mL NGF (R&D Systems), and 0.25 pM Arabinose Cytoside.

### 2.6. Co-Culture of Trigeminal Ganglia and AB10 Cells in Microfluidic Devices

Trigeminal ganglia were placed through the punched hole into the neuronal chamber of the microfluidic device [36,37]. Ganglia were cultured alone for 5–7 days, until axons were projected into the stem cells chamber. Cells were then added to the co-culture system. Of AB10 cells 20 μL was added directly in the stem cell chambers at a density of 1 × 10^4^ cells/chamber. Cells were left to attach for 1 h and 30 min; then 300 μL of the appropriate medium were added to all chambers. Co-cultures were maintained for four days. At the end of the culture period, the medium was removed, and all chambers were washed once with 300 μL PBS for 10 min. Afterwards, the samples were fixed in paraformaldehyde 4% (PFA 4%), 300 μL/chamber, for 15 min. Chambers were then washed again with 300 μL PBS/chamber for 10 min and were stored at 4 °C in PBS.

### 2.7. Immunofluorescent Staining in Microfluidic Devices

Fixed samples were permeabilized by incubating them in 1% Triton/PBS for ten minutes. Non-specific sites were blocked with PBS, 0.1% Triton, and 1% bovine serum albumin (BSA) for 30 min at RT, 90–100 μL per chamber. Co-cultures were incubated overnight with the following primary antibodies (4 °C): mouse IgG1 α-Neurofilament (α-NF) antibody (1:100, Hybridoma Bank, Iowa City, USA), rabbit pAb anti-Synapsin I (1:100, Abcam, ab64581) diluted in PBS supplemented with 1% Bovine Serum Albumin (BSA). For negative controls, primary antibody was omitted. The samples were washed three times (5 min each) with PBS + 0.5% Tween. The co-cultures were then incubated overnight with the following secondary antibodies (50–60 μL per chamber): Alexa-Fluor647 conjugated anti-Rabbit (A32795, ThermoFisher), Alexa-fluor568 conjugated anti-mouse (A-11004, ThermoFisher) diluted 1:250 in 1% BSA/PBS. Co-cultures were then washed three times (5 min each) with PBS and successively stained with Phalloidin (A12379, ThermoFisher). The samples were then incubated with 4′,6-Diamidino-2-Phenylindole (DAPI) for ten minutes at RT. Samples were finally washed twice (5 min each) with PBS. Microfluidic devices were then removed from the 6-wells plates and mounted on slides. Co-cultures were imaged with a Leica SP8 Confocal Laser Scanning Microscope (CLSM). 3D reconstructions were realized with Fiji/ImageJ [39].

## 3. Results

### 3.1. Histological and Molecular Characterization of Human Ameloblastomas

We first characterized the histological appearance of human ameloblastoma biopsies. Ameloblastomas were characterized by anastomosing strands of epithelial, preameloblast-like basal cells, which constitute an important part of the cancer epithelium (Figure 1A,B). Epithelial cells expressed the dental epithelium marker Keratin 14 (Krt14) in an inhomogeneous manner, as highly Krt14-expressing, follicular areas were alternated with Krt14-negative regions in the ameloblastoma epithelium (Figure 1C,D). In specific regions, the epithelial expression of Krt14 would fade into the underlying mesenchymal tissue, suggesting the presence of epithelial–mesenchymal transformation (EMT) events (Figure 1D, red arrowheads). To verify the existence of cells undergoing EMT, we performed double immunofluorescent staining against wide spectrum keratins (pKrt), which mark epithelial cells, and the mesenchymal marker Vimentin (Figure 1F–H). The staining revealed the existence of double pKrt^+^ Vimentin^+^ cells at the interface between the epithelium and the stroma (Figure 1G; yellow arrowheads), as well as within the stroma (Figure 1H; yellow arrowheads).

We further investigated whether the analyzed ameloblastomas were vascularized. Neo-angiogenesis and the subsequent increased supply of nutrients are fundamental hallmarks of cancers [40,41], and vessel growth is often coupled with innervation [42]. Immunofluorescent staining against the endothelial cells’ marker CD31 showed that these ameloblastomas were indeed vascularized (Figure 1I). Blood vessels could be detected within the Vimentin^+^ stroma (Figure 1J) as well as in close proximity to the Vimentin^−^ ameloblastoma epithelium (Figure 1K,L). Vessels lined the epithelium at the interface with the stroma (Figure 1K) and also entered fully epithelial regions (Figure 1L).

We then assessed the expression of the dental epithelial stem cell markers BMI1 and SOX2 in human ameloblastoma biopsies. The malignant epithelium was characterized by the vast expression of both BMI1 (Figure 1M,N) and SOX2 (Figure 1O,P), showing that these cells could maintain, or reactivate, the expression of genes that promote self-renewal. BMI1 expression was particularly enriched in the epithelial cells located in proximity to the stroma (Figure 1M,N), while SOX2 expression appeared more scattered and more pronounced within the inner epithelial layers (Figure 1O,P).

Our immunostaining was not quantitative, as we did not compare the expression of the analyzed molecules between different ameloblastoma subtypes.

We then characterized the expression of Notch receptors and ligands in these specimens. Ameloblastomas expressed high levels of NOTCH2 (Figure 2A–D). NOTCH2 expression was notably concentrated at the interface between the malignant epithelium and the stroma (Figure 2B–D), and it lined the whole extension of the cancers observed. However, the thickness of the NOTCH2-expressing layer varied considerably. The field of NOTCH2 expression was generally broader in correspondence with epithelial strands invaginating the underlying stroma, while it was thinner in association with linear epithelium/stromal interfaces (Figure 2A,C). Ameloblastomas expressed detectable levels of NOTCH3 (Figure 2E–H). NOTCH3 expression partially overlapped with that of NOTCH2, as it was detected at the interface between the ameloblastoma epithelium and the underlying stroma (Figure 2E,F). However, expression of NOTCH3 was more pronounced on the membranes of epithelial cells facing the stroma (Figure 2F) when compared to NOTCH2 expression. NOTCH3 was also detected within the ameloblastoma epithelium, where NOTCH3 expressing regions were irregularly alternated with NOTCH3 negative areas (Figure 2G,H).

Expression of NOTCH2 and NOTCH3 was associated with expression of the ligands JAG1, DLL1, and DLL4 in the ameloblastoma epithelium (Figure 2I–P). JAG1 was expressed throughout the ameloblastoma epithelium (Figure 2I). Its expression appeared generally higher in the epithelial cells directly facing the stroma, while low levels of JAG1 expression were detected mostly in the inner epithelial layers (Figure 2I,J,L). JAG1 was uniformly expressed in epithelial islets located within the stroma (Figure 2K). DLL1 was expressed variably by the outer and the inner layers of the epithelial strands (Figure 2M,N). DLL4 was mostly expressed on the more superficial layer of the epithelium (Figure 2O,P), i.e., the cell layers in closer contact with the NOTCH2-expression field and the stroma (Figure 2C,D).

### 3.2. Innervation and Vascularization of Human Ameloblastomas

We then investigated the possibility that ameloblastomas are innervated. For this purpose, we first performed immunohistochemical staining against the pan-neuronal marker βIII-Tubulin. βIII-Tubulin marked many elongated structures within the mesenchyme that projected towards the cancerous epithelium (Figure 3A–D). βIII-Tubulin signal could be detected in close proximity to the epithelium, as well as within epithelial layers (Figure 3C,D).

We then performed immunohistochemistry against PGP9.5, also known as ubiquitin carboxyl-terminal hydrolase-1 (UCH-L1), a marker commonly used to stain neurons and neuroendocrine cells [43] (Figure 3E,F). PGP 9.5 immunoreactivity was detected in a subset of axonal structures within the stroma (Figure 3G,H—red arrowheads) often associated with blood vessels, which represented a small proportion of those marked by βIII-Tubulin. PGP 9.5 also marked scattered clusters of cells within the malignant epithelium (Figure 3H). The latter pattern could indicate invasion of nerve fibers within the ameloblastoma epithelium or expression of PGP9.5 by ameloblastoma cells, as already reported in other cancer types [44].

We performed double immunofluorescent staining against βIII-Tubulin and the mesenchymal cells marker to corroborate the identity and the localization of the observed βIII-Tubulin-expressing cells (Figure 3I–L). Indeed, double staining revealed the presence of many βIII-Tubulin^+^/Vimentin^−^ cells in close contact with the ameloblastoma epithelium (Figure 3J,L; white arrowheads), thus supporting a neuronal, non-mesenchymal identity for the observed βIII-Tubulin^+^ cells. We assessed the exact relative localization of afferent nerve fibers and malignant ameloblastoma epithelium by performing double immunofluorescent staining against the pan-neuronal marker neurofilament and keratins (Figure 3M–P). Neurofilament^+^ nerve fibers (white arrowheads) were detected well aligned within the stroma. From here they extended along the main axes of the epithelial strands (Figure 3M,N). Axons were detected adjacent to the ameloblastoma epithelium in many locations (Figure 3O). Direct contacts were observed between the nerve fibers and the epithelium, especially in correspondence to epithelial strands projecting into the stroma (Figure 3P).

We then assessed whether the axonal growth towards the tumor epithelium could be due to the expression of neurotrophins, which are key promoters of tissue innervation. Using immunofluorescent staining, we detected expression of the neurotrophins NGF (Nerve Growth Factor) and BDNF (Brain Derived Neurotrophic Factor), as well as their receptors P75/NGFR, TRKA and TRKB, in human ameloblastoma biopsies (Figure 4). NGF was strongly expressed in discrete regions within the cancer stroma (Figure 4A–C). The NGF signal could also be detected within the ameloblastoma epithelium, in particular in cells located in proximity to the underlying stroma (Figure 4C–D; yellow arrowheads). BDNF was expressed mostly in the ameloblastoma epithelium (Figure 4E,F). Its expression appeared higher than that of NGF and was distributed in patches throughout the epithelium without an obvious correlation with the distance from the stroma (Figure 4E,F; white arrowheads). Immunofluorescent staining also revealed expression of the neurotrophin receptors P75/NGFR, TRKA, and TRKB. P75/NGFR was expressed at high levels throughout the ameloblastoma epithelium (Figure 4A), as well as in stromal regions associated to NGF expression (Figure 4B). TRKA displayed a similar expression pattern, as it was detected in the epithelium as well as in close proximity to NGF-expressing clusters within the stroma (Figure 4C,D). TRKB expression was high in highly clustered regions located within the stroma (Figure 4E), while it was very low or undetectable within the ameloblastoma epithelium (Figure 4F).

### 3.3. Co-Culture of Trigeminal Ganglia and Human Ameloblastoma Cells in a Microfluidics Co-Culture System

We then proceeded to investigate whether isolated human ameloblastoma cells possess neurotrophic properties. For this purpose, we exploited the AB10 ameloblastoma cell line [6].

AB10 cells expressed high levels of the epithelial marker *CDH1*, as well as detectable levels of the dental epithelial stem cell markers *BMI1*, *SOX2*, and *TP63* (Figure 5A), thus preserving the expression profile observed in vivo (Figure 1). We also detected expression of mesenchymal markers such as *ACTA2* and *FN1* (coding for smooth muscle actin and fibronectin, respectively; (Figure 5A). This suggests that AB10 cells are in an intermediate EMT (Epithelial–Mesenchymal Transition) state. AB10 cells also expressed *NOTCH1-3*, Notch ligands *DLL1* and *JAG2*, and the *NOTCH* target gene *HES1* (Figure 5B). AB10 cells expressed detectable levels of *NGF* and *BDNF* (Figure 5C), in accordance with what we observed in vivo (Figure 4).

AB10 cells cultured in adhesive conditions formed an epithelial-like monolayer (Figure 5D,E), as previously reported [6]. When cultured in sphere-forming conditions [34], AB10 cells could generate spheres with high efficiency (Figure 5F,G). This behavior, together with the expression of key dental epithelial stem cell markers, suggests that AB10 cells continued to preserve stem cell properties even after extensive culturing.

We then probed the neurotrophic properties of ameloblastoma cells in a microfluidic co-culture system (Figure 6A) [6,36,37]. In this system we co-cultured whole trigeminal ganglia and AB10 cells (Figure 6A). Trigeminal axons invaded the AB10-containing culture chamber and displayed a highly directional growth towards AB10 cells (Figure 6B,C). Long axons could be observed pointing towards isolated or grouped AB10 cells (Figure 6C,D; black arrowheads). Once the axons reached their target, they formed extensive networks characterized by abundant branching (Figure 6E,F).

We characterized in greater detail the innervation of AB10 cells by high-resolution confocal microscopy. Neurofilament^+^ trigeminal axons crossed the microgrooves and contacted AB10 cells (Figure 7A). Axons established extensive 3D networks in correspondence of AB10 cells, contacting them both at their adherent inferior region and at their superior surface (Figure 7B,C; Appendix A). Axon branching was often observed in correspondence of AB10 cells innervation (Figure 7D). Multiple contacts were established between the trigeminal axons and the target AB10 cells (Figure 7E). To determine whether synaptic activity could be associated with these contacts, we performed immunostaining against Synapsin I, a neuronal phosphoprotein that coats synaptic vesicles and modulates neurotransmitter release. We observed that the axonal terminals contacting AB10 cells were characterized by evident accumulation of Synapsin I (Figure 7F–H).

## 4. Discussion

Ameloblastoma is a slow growing but locally infiltrative tumor of the oral cavity, and its pathogenesis is very poorly understood. Very few studies concerning the molecular and cellular bases of this neoplasia have been conducted, thus limiting pharmacological interventions to a strictly reduced number of options [3,45]. As a result, most experimental therapeutic approaches are aimed at targeting the MAPK pathway [7,45] and Hedgehog signaling [5,7]. 

A common feature of cancers is the persistence or re-activation of stem cell-specific transcriptional and signaling networks in a subset of malignant cells [46]. These Cancer Stem Cells (CSCs) give rise to shorter-lived malignant cells that make up the bulk of the tumor. At the same time, they are able to sustain long-term tumor growth thanks to their self-renewal ability and their increased resistance to chemotherapy and radiotherapy [46,47]. Moreover, CSCs are thought to be primarily responsible for tumor relapse [46,47]. Although solid evidence supports such a hierarchical organization in many tumor types, it is not clear whether this would apply to all cancer types [48]. Moreover, the proportion of CSCs within a cancer is extremely variable [48]. In our study we observed a widespread expression of the dental epithelial stem cell markers SOX2 and BMI1 within ameloblastoma. This suggests that a major proportion of the cells composing these tumors present CSC-like properties. This is further supported by the observed expression of Notch ligands and receptors. Notch signaling is a fundamental player in the establishment of stem cell fate and behavior in development, regeneration and disease [12,13,14,49]. Notch signaling was shown to be necessary for the maintenance of cancer stem cells in different types of tumors [18,50,51,52]. As such, it is the target of numerous therapeutic antineoplastic approaches [15,18,53]. Notch ligands and receptors are expressed in the dental epithelium in adjacent layers during tooth development, modulating cytodifferentiation [35,54,55,56,57,58,59]. Their expression is low or absent in adult dental tissues but is reactivated upon injury [60,61,62,63,64]. Here, we showed that NOTCH2 and NOTCH3, but not NOTCH1 (data not shown), are expressed at the interface between the ameloblastoma epithelium and the underlying stroma, which is a location pivotal for the acquisition of an invasive phenotype [65]. The presence of Notch ligands such as JAG1, DLL1, and DLL4 in malignant epithelial cells adjacent to the NOTCH2^+^ NOTCH3^+^ interface suggests a possible involvement of Notch signaling in the establishment of ameloblastoma invasiveness and in the interaction of malignant epithelial cells with the underlying stroma [18,53,66]. We observed that NOTCH3, JAG1, DLL1, and DLL4 are also expressed within the ameloblastoma tumor mass. As their expression is associated with that of the dental epithelial stem cell markers BMI1 and SOX2, it is reasonable to hypothesize that Notch ligands and receptors might contribute to the maintenance of stemness of ameloblastoma cells. Expression of Notch receptors and ligands has been reported in previous studies with contrasting results [9,10,11]. Some studies have reported the expression of NOTCH1, 2, and 3, as well as their ligands DLL1, DLL4, and JAG1, in all ameloblastomas analyzed [9]. On the other hand, other studies have reported absent expression of NOTCH2 and high expression of NOTCH4 [11,67]. While these discrepancies do not appear to correlate with the subclasses of ameloblastoma analyzed, all studies still detected abundant expression of different combinations of Notch ligands and receptors in human ameloblastomas. Taken together, these results indicate that Notch signaling activation is strongly associated with ameloblastoma pathogenesis and might play an important role in its course.

Tumorigenesis is driven by a plethora of factors, and it is a combination of these factors that determines the mode of cancer onset, progression, and ultimately their prognosis. Among these factors, innervation has been long neglected. More and more studies support the importance of nerve-derived signals in organ development, homeostasis, regeneration, and in pathological conditions [20,68]. It is well known that most cancers are associated with pain sensation [69], and different studies have reported the expression of neurotrophic factors in many cancer types [70,71,72,73,74]. Only recently a few studies have started shedding light on the fundamental role of innervation for cancer onset and progression. It was indeed shown that nerve-derived signals promote the progression of different types of tumor, including gastric, pancreatic, breast, and oral cancers [20,21,23,25,26,29,75]. For instance, in prostate cancer, recruitment of nerve fibers to cancer tissue increases tumor proliferation and is associated with a higher risk of recurrence and metastasis [21]. Similarly, denervation studies carried out on mouse cancer models suggest that nerves directly enhance and fuel tumor progression [27,76]. These studies thus strongly indicate that the nervous system is an active participant in carcinogenesis.

Our results show that human ameloblastomas are innervated, and that the afferent nerve fibers enter in close contact with the tumor epithelium. We also showed that ameloblastoma malignant cells themselves exert strong neurotrophic effects. Nerve-derived signals could therefore be fundamental for ameloblastoma pathogenesis, as already proven for other cancers [21,23,26,27,77,78]. Innervation might indeed directly modulate ameloblastoma proliferation and invasiveness. We have shown that the ameloblastoma epithelium expresses detectable levels of the neurotrophins NGF and BDNF, as well as the neurotrophin receptors TRKA, TRKB, and P75/NGFR. Both NGF and BDNF have already been shown to promote tumor innervation in other tissues [29,72,74,79]. In addition to promoting cancer innervation, neurotrophin-mediated signaling has been shown to directly drive tumor onset and progression [72,74,80,81,82]. The neurotrophin receptors TRKA, TRKB, and P75/NGFR are expressed by malignant cells in many tumor types, where they inhibit key tumor suppressor mechanisms [82] or directly promote cancer cell proliferation and invasiveness [71,74,80,81,83]. Our results show that ameloblastomas expressed both neurotrophins and neurotrophin receptors. Their expression thus leads to neuroattraction and, at the same time, could induce activation of neurotrophin signaling within cancer cells themselves. Neurotrophins and their receptors are attracting increasing attention as therapeutic targets in many cancer types, as pharmacological interventions against this signaling axis have the potential to not only target cancer cells directly, but also to inhibit neurogenesis and its stimulatory effects on cancer progression and pain [73,84]. Indeed neural regulation represents an emerging targetable pathway for the treatment of cancer [75,76,78,85]. As cancers develop, axonal networks form in and around the tumor stroma, providing signals that modulate cancer progression [21,75]. Clinical data show that cancer patients treated with β-blockade, which impairs sympathetic nerve functions, have improved survival, supporting the role of nerve activity in cancer progression [24,75,86].

Taken together, our results show that ameloblastomas exhibit stem cell potential, express NGF, BDNF, and neurotrophin receptors, and create contacts with trigeminal neurons that might play key roles in the onset and progression of these tumors. The neural-cancer axis could thus constitutes a promising target for the treatment of ameloblastoma.

## Figures and Tables

**Figure 1 cells-09-00644-f001:**
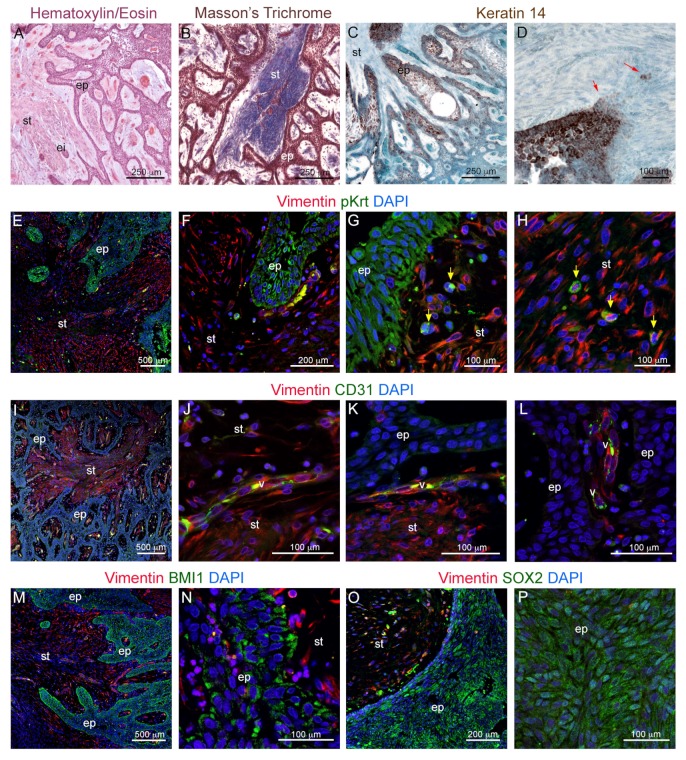
(**A**) Hematoxylin eosin staining of plexiform ameloblastoma. (**B**) Trichrome Masson’s Staining of plexiform ameloblastoma. (**C**,**D**) Immunohistochemical staining showing Keratin 14 (Krt14) distribution in ameloblastomas. Red arrowheads indicate Krt14^+^ cells invading the stroma. (**E**–**H**) Double immunofluorescent staining against the epithelial cell marker wide-spectrum-Keratin (pKrt; green color) and the mesenchymal cell marker Vimentin (red color). (**E**) Overview of epithelial and mesenchymal cells distribution. (**F**,**G**) Higher magnification of the epithelium/stroma interface. (**H**) Higher magnification of the stromal region, showing infiltration of pKrt^+^Vimentin^+^ cells. Blue color: DAPI. Yellow arrowheads indicate pKrt^+^ cells invading the stroma. (**I**–**L**) Double immunofluorescent staining against Vimentin (red color) and CD31 (green color). Blue color: DAPI. (**M**,**N**) Double immunofluorescent staining against BMI1 (green color) and Vimentin (red color). Blue color: DAPI. (**O**,**P**) Double immunofluorescent staining against SOX2 (green color) and Vimentin (red color). Blue color: DAPI. Abbreviations: ei, epithelial islet; ep, epithelium; Krt14, Keratin 14; pKrt, pan-Keratin; st, stroma; v, vessel. Scale bars: (**A**–**C**): 250 μm; (**D**,**G**,**H**,**J**–**L**,**N**,**P**): 100 μm; (**E**,**I**,**M**): 500 μm; (**F**,**O**): 200 μm.

**Figure 2 cells-09-00644-f002:**
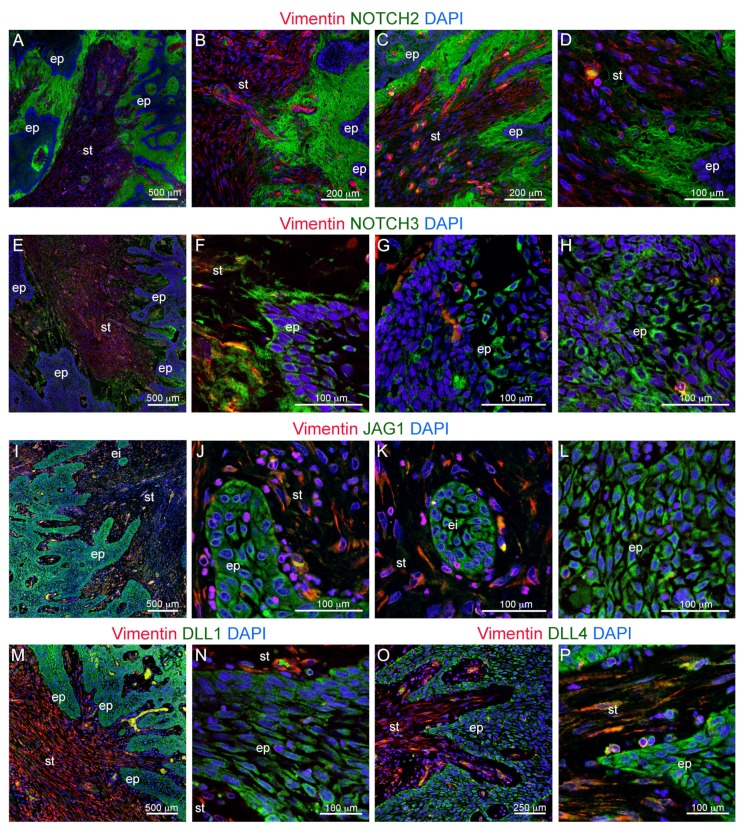
(**A**–**D**) Double immunofluorescent staining against NOTCH2 (green color) and Vimentin (red color). (**E**–**H**) Double immunofluorescent staining against NOTCH3 (green color) and Vimentin (red color). Blue color: DAPI. (**I**–**L**) Double immunofluorescent staining against JAG1 (green color) and Vimentin (red color). (**M**,**N**) Double immunofluorescent staining against DLL1 (green color) and Vimentin (red color). (**O**,**P**) Double immunofluorescent staining against DLL4 (green color) and Vimentin (red color). Blue color: DAPI. Abbreviations: ei, epithelial islet; ep, epithelium; st, stroma. Scale bars: (**A**,**E**,**I**,**M**): 500 μm; (**B**,**C**): 200 μm; (**D**,**F**–**H**,**J**–**L**,**N**,**P**): 100 μm; (**O**): 250 μm.

**Figure 3 cells-09-00644-f003:**
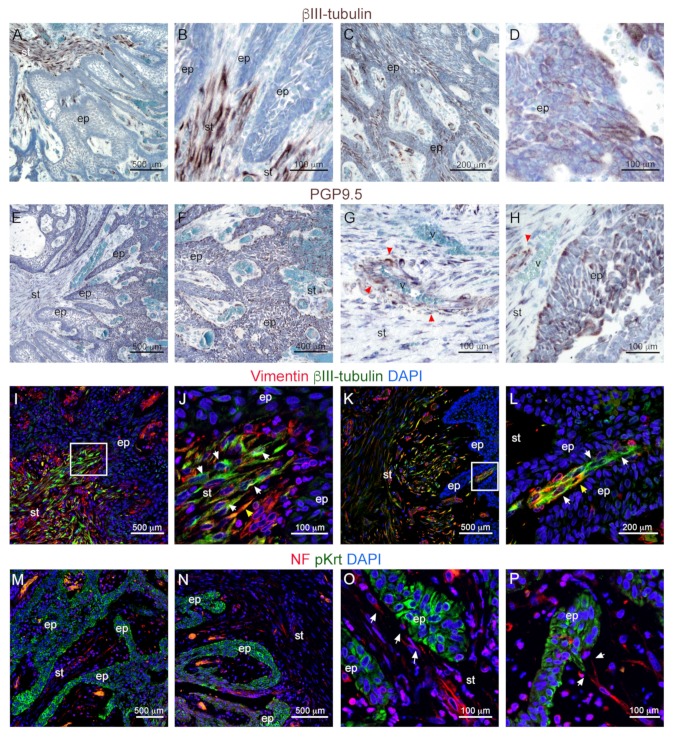
Innervation of ameloblastomas. (**A**–**H**) Immunohistochemistry against (**A**–**D**) βIII-Tubulin (brown color) and (**E**,**F**) PGP9.5 on ameloblastomas. Counterstaining with Toluidine Blue (blue color). Red arrowheads: neurons associated with vessels. (**I**–**L**) Double immunofluorescent staining against βIII-Tubulin (green color) and Vimentin (red color). Blue color: DAPI. White arrowheads: βIII-Tubulin^+^ Vimentin^−^ nerve fibers; yellow arrowheads: double βIII-Tubulin^+^ Vimentin^+^ mesenchymal cells. White boxes in (**I**) and (**K**) indicate the areas shown in (**J**) and (**L**), respectively. (**M**–**P**) Double immunofluorescent staining against neurofilament (red color) and keratins (green color). Blue color: DAPI. White arrowheads: nerve fibers in close contact with the ameloblastoma epithelium. Abbreviations: ep, epithelium; NF, neurofilament; pKrt, pan-Keratins; st, stroma; v, vessels. Scale bars: (**A**,**E**,**I**,**K**,**M**,**N**): 500 μm; (**B**,**D**,**G**,**H**,**J**,**O**,**P**): 100 μm; (**C**,**L**): 200 μm; (**F**): 400 μm.

**Figure 4 cells-09-00644-f004:**
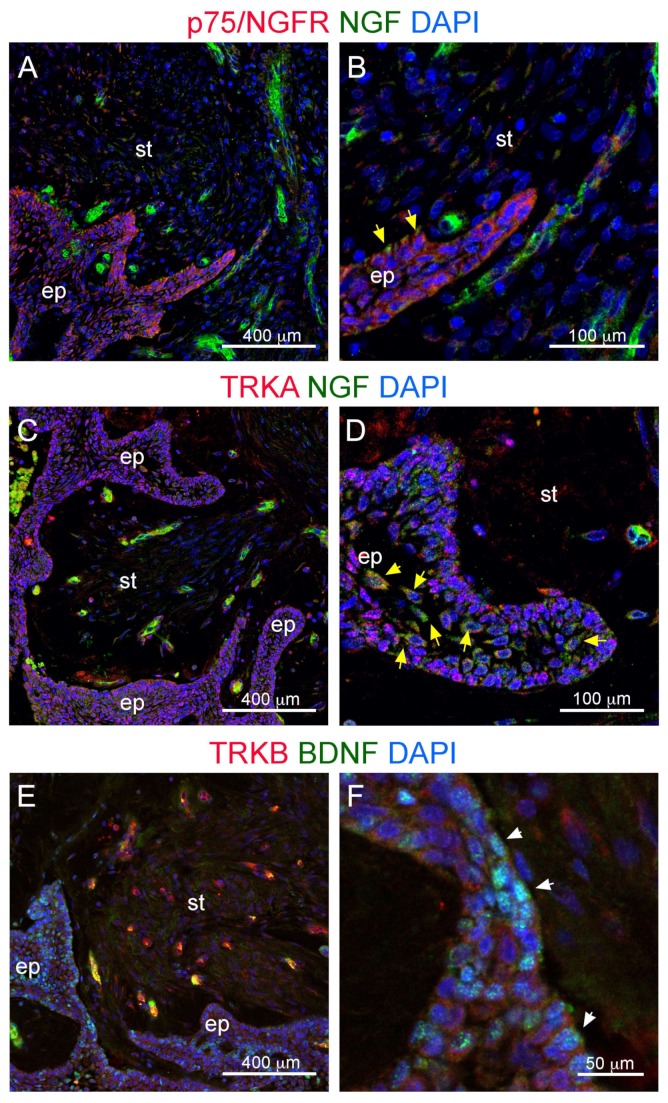
Double immunofluorescent staining against neurotrophins and their receptors. (**A**,**B**) Immunofluorescent staining against P75/NGFR (red color) and NGF (green color). (**C**,**D**) Immunofluorescent staining against TRKA (red color) and NGF (green color). (**E**,**F**) Immunofluorescent staining against TRKB (red color) and BDNF (green color). Blue color: DAPI. Yellow arrowheads (**B**,**D**) and white arrowheads (**F**) indicate, respectively, NGF expressing- and BDNF expressing-cells within the ameloblastoma epithelium. Abbreviations: BNDF, Brain Derived Neurotrophic Factor; ep, epithelium; NGF, Nerve Growth Factor; st, stroma. Scale bars: (**A**,**C**,**E**): 400 μm; (**B**,**D**): 100 μm; (**F**): 50 μm.

**Figure 5 cells-09-00644-f005:**
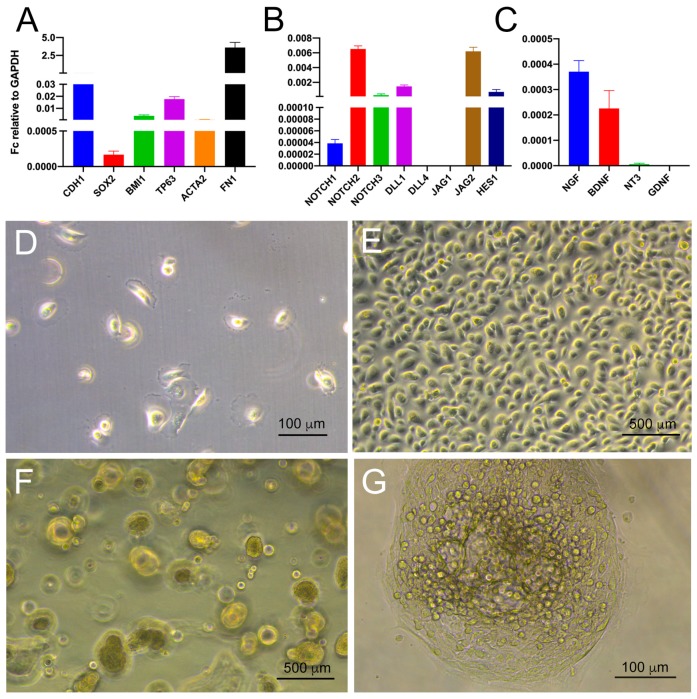
Characterization of AB10 cells. (**A**–**C**) Real time PCR analysis of the expression of genes coding for epithelial and mesenchymal markers (**A**), for NOTCH pathway components (**B**), and for neurotrophins (**C**). (**D**–**G**) Brightfield images showing AB10 cells grown in adhesion (**D**,**E**) and 3D/sphere-forming conditions (**F**,**G**). Scale bar: (**D**,**G**): 100 μm; (**E**,**F**): 500 μm.

**Figure 6 cells-09-00644-f006:**
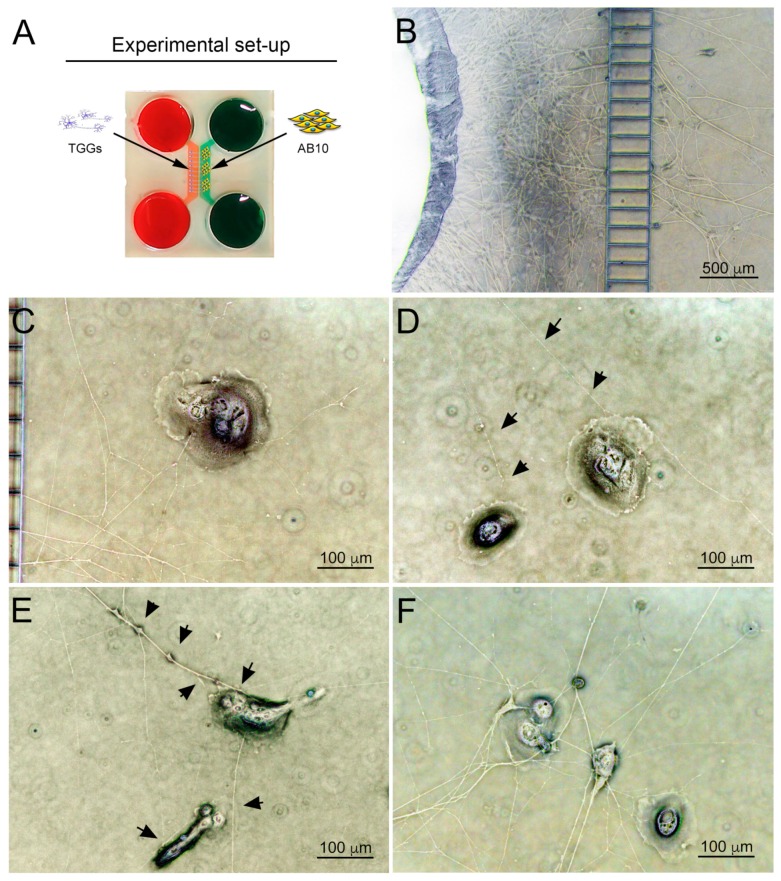
Co-culture of trigeminal ganglia and human AB10 ameloblastoma cells. (**A**) Schematic representation of the experimental set-up. (**B**) Overview of axons crossing the microgrooves and entering the AB10-containing culture chamber. (**C**–**F**) Brightfield images showing directional growth of axons towards AB10 cells at different distances from the chamber’s borders. Black arrowheads highlight axons growing towards AB10 cells. Scale bars: (**B**): 500 µm; (**C**–**E**): 100 µm.

**Figure 7 cells-09-00644-f007:**
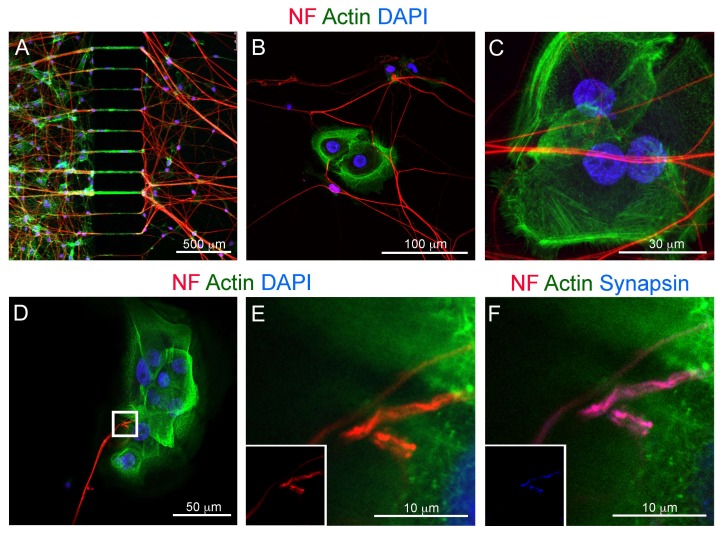
Confocal microscopy imaging of co-cultures of trigeminal ganglia and AB10 ameloblastoma cells. (**A**) Double immunofluorescent staining against neurofilament (red color) and actin (phalloidin; green color) showing passage of axons from the neuronal culture chamber to the AB10-containing culture chamber. Blue color: DAPI. (**B**) Double immunofluorescent staining against neurofilament (red color) and actin (phalloidin; green color) showing axonal networks formed by trigeminal neurons in correspondence of AB10 cells. Blue color: DAPI. (**C**) 3D projection of the innervation of a cluster of AB10 cells. Neurofilament (red color) marks axons and phalloidin (against actin; green color) marks AB10 cells cytoskeleton. Blue color: DAPI. (**D**) Double immunofluorescent staining against neurofilament (red color) and actin (phalloidin; green color) showing axonal branching and establishment of multiple connections upon contact of trigeminal axons with AB10 cells. Blue color: DAPI. (**E**) Higher magnification of D (white box in D indicates the region showed in E). Insert shows single channel immunofluorescent staining against neurofilament. (**F**) Triple immunofluorescent staining against neurofilament (red color), actin (green color) and Synapsin I (blue color) showing accumulation of synapsin at the sites of axon/AB10 cells contact. Insert shows single channel immunofluorescent staining against Synapsin I. Scale bars: (**A**): 500 µm; (**B**): 100 µm; (**C**): 30 µm; (**D**): 50 µm; (**E**): 10 µm.

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
