# Peer review of "Ameloblastomas Exhibit Stem Cell Potential, Possess Neurotrophic Properties, and Establish Connections with Trigeminal Neurons"

_cells, 2020, doi:10.3390/cells9030644_

Round 1
Reviewer 1 Report
Dear Authors,
Your detailed description of molecular characteristics and neurotrophic properties of human ameloblastomas will be appreciated by the reader.
In detail, the results regarding:
- the iper-expression of the integrinic Notch receptor,
- the relevant role of innervation confirmed by immuno fluorescence besides the neurotrophic properties,
- the high vascularization detected by the CD 31
support your sound conclusions about the ameloblastoma's biologic behaviour.
Your work can improve the in deep understanding of molecular and cellular basics of ameloblastomas and could represent a starting point for new targeted therapies.
For these reasons I strongly recommend the publication of your appreciated paper.
Author Response
Dear Authors,
Your detailed description of molecular characteristics and neurotrophic properties of human ameloblastomas will be appreciated by the reader.
In detail, the results regarding:
- the iper-expression of the integrinic Notch receptor,
- the relevant role of innervation confirmed by immuno fluorescence besides the neurotrophic properties,
- the high vascularization detected by the CD 31
support your sound conclusions about the ameloblastoma's biologic behaviour.
Your work can improve the in deep understanding of molecular and cellular basics of ameloblastomas and could represent a starting point for new targeted therapies.
For these reasons I strongly recommend the publication of your appreciated paper.
---------------
We thank the reviewer for the positive feedback.
Reviewer 2 Report
The authors Pagella and colleagues performed a study analyzing the exhibitions of stem cell potential, neurotrophic properties and the establishment with trigeminal neurons by ameloblastomas. The study is well performed and the manuscript is well written. The topic of the study is of significant interest in the field of dentistry, head and neck surgery, -oncology and -pathology.
The abstract should focus on the results of the study in more detail. Please change.
In the chapter "Introduction", the authors should add some information concerning the current WHO classification of odontogenic tumors with respect to the ameloblastoma. Next, the different clinical and pathological types of ameloblastomas should be presented.
Materials and Methods: Which types of ameloblastomas according the WHO classification were used for the study? Did you use soft-tissue or bone samples for the study? How many samples were used in total? Further, it might be of interest to give some basic information about the patients the samples were harvested from, such as gender, age, location etc. How many mice were used in total? Please add detailed information about the mice including housing, medications, narcotic mixtures, food etc. Was the animal care in accordance with the ARRIVE guidelines? Overall, the authors should review the chapter "Materials and Methods" in detail, also with respect to the experimental numbers, the numbers of samples for immunofluorescent staining, and the numbers of samples or experiments that were included to the analysis. Where there any controls, such as positive and negative controls?
What kind of statistical analysis was used in the study? The results part is nitty-gritty. The count of the figures should be reduced. For the immunofluorescent staining, no quantitative analysis including a further statistical analysis has been performed? This might be reasonable, please add.
The chapter "Discussion" should include possible clinical consequences of these findings for the clinician.
Author Response
The authors Pagella and colleagues performed a study analyzing the exhibitions of stem cell potential, neurotrophic properties and the establishment with trigeminal neurons by ameloblastomas. The study is well performed and the manuscript is well written. The topic of the study is of significant interest in the field of dentistry, head and neck surgery, -oncology and -pathology.
We thank the reviewer for the positive feedback about the importance of our study
The abstract should focus on the results of the study in more detail. Please change.
We edited the abstract in accordance with the comment of the reviewer.
In the chapter "Introduction", the authors should add some information concerning the current WHO classification of odontogenic tumors with respect to the ameloblastoma. Next, the different clinical and pathological types of ameloblastomas should be presented.
The Introduction has been modified accordingly.
Materials and Methods: Which types of ameloblastomas according the WHO classification were used for the study? Did you use soft-tissue or bone samples for the study? How many samples were used in total? Further, it might be of interest to give some basic information about the patients the samples were harvested from, such as gender, age, location etc. How many mice were used in total? Please add detailed information about the mice including housing, medications, narcotic mixtures, food etc. Was the animal care in accordance with the ARRIVE guidelines? Overall, the authors should review the chapter "Materials and Methods" in detail, also with respect to the experimental numbers, the numbers of samples for immunofluorescent staining, and the numbers of samples or experiments that were included to the analysis. Where there any controls, such as positive and negative controls?
All these issues have been addressed and included in the revised manuscript.
What kind of statistical analysis was used in the study? The results part is nitty-gritty. The count of the figures should be reduced. For the immunofluorescent staining, no quantitative analysis including a further statistical analysis has been performed? This might be reasonable, please add.
Our study is not quantitative, i.e. we did not compare the expression of the analyzed molecules between different ameloblastoma subtypes. This has been specified in our revised manuscript.
The figures have been merged and reorganized, and the total count reduced (to 7 panels).
The chapter "Discussion" should include possible clinical consequences of these findings for the clinician.
The issue has been addressed in the revised version of the manuscript.
Reviewer 3 Report
Interesting and well written article.
Methods are appropriate. Results are clearly presented.
Author Response
Interesting and well written article.
Methods are appropriate. Results are clearly presented.
------------
We thank the reviewer for the very positive feedback.
Round 2
Reviewer 2 Report
The authors commented and answer all comments and questions in detail. I have no further questions or comments. I would like to recommend to accept the manuscript for publication. Thank you for your fine contribution.